# Effects of an Online Supervised Exercise Training in Children with Obesity during the COVID-19 Pandemic

**DOI:** 10.3390/ijerph19159421

**Published:** 2022-08-01

**Authors:** Matteo Vandoni, Vittoria Carnevale Pellino, Alessandro Gatti, Daniela Lucini, Savina Mannarino, Cristiana Larizza, Virginia Rossi, Valeria Tranfaglia, Agnese Pirazzi, Valentina Biino, Gianvincenzo Zuccotti, Valeria Calcaterra

**Affiliations:** 1Laboratory of Adapted Motor Activity (LAMA), Department of Public Health, Experimental Medicine and Forensic Science, University of Pavia, 27100 Pavia, Italy; vittoria.carnevalepellino@unipv.it (V.C.P.); alessandro.gatti08@universitadipavia.it (A.G.); agnese.pirazzi01@universitadipavia.it (A.P.); 2Department of Industrial Engineering, University of Rome Tor Vergata, 00133 Rome, Italy; valeria.calcaterra@unipv.it; 3BIOMETRA Department, University of Milan, 20129 Milan, Italy; daniela.lucini@unimi.it; 4Exercise Medicine Unit, Istituto Auxologico Italiano IRCCS, 20135 Milan, Italy; 5Pediatric Department, “Vittore Buzzi” Children’s Hospital, 20154 Milan, Italy; savina.mannarino57@gmail.com (S.M.); virginia.rossi@unimi.it (V.R.); valeria.tranfaglia@unimi.it (V.T.); gianvincenzo.zuccotti@unimi.it (G.Z.); 6Department of Electrical, Computer and Biomedical Engineering, University of Pavia, 27100 Pavia, Italy; cristiana.larizza@unipv.it; 7Department of Human Sciences, University of Verona, 37129 Verona, Italy; valentina.biino@univr.it; 8School University of Medicine and Surgery, University of Verona, 37129 Verona, Italy; 9Department of Biomedical and Clinical Science, Università degli Studi di Milano, 20157 Milan, Italy; 10Department of Internal Medicine, University of Pavia, 27100 Pavia, Italy

**Keywords:** online exercise, physical fitness, childhood obesity, childhood exercise intervention

## Abstract

COVID-19 restrictions have dramatically reduced the active lifestyle and physical activity (PA) levels in the whole population, a situation that can contribute to weight gain and to develop obesity. To improve physical fitness (PF) in children with obesity during COVID-19 restrictions, sport specialists started to deliver physical training through tele-exercise. For these reasons, the aim of this study was to evaluate the effects of a 12-week online supervised training program in children with obesity on different PF components and PA levels. We enrolled a total of 40 Caucasian children (9 F/31 M; aged 11 ± 1.9 years) with obesity. The data collection consisted of a series of anthropometric measures, the PAQ-C questionnaire, and PF tests, valid and reliable tools to assess PF in children. We used a Wilcoxon’s *t*-test and a Student’s *t*-test, as appropriate, to assess the differences before and after the training protocol. A total of 37 patients completed the training protocol and were considered in the analysis. Our results show an improvement in all the PF tests, a reduction in the BMI z-score, the waist circumference, and in the waist-to-height ratio, and an increased PA level. In conclusion, the results of our study show that an online supervised training program is effective to promote PA, improving PF and reducing the BMI z-score in children with obesity.

## 1. Introduction

The COVID-19 pandemic has spread all over the world and the disease causes severe damages such as cough, fever, respiratory, and cardiovascular distress. For these reasons, the authorities implemented special measures, such as the lockdown, to limit social interaction and travels, including the closures of sports and training facilities. Even if these restrictions have been shown to be effective to reduce the spread of the virus, the daily routine of everyone changed and, consequently, the active lifestyle and physical activity (PA) levels dramatically decreased in the whole population [1,2,3,4]. Moreover, the increase of sedentariness and an unhealthy diet contributed to weight gain and to develop obesity (OB), especially in the young population. Before the COVID-19 pandemic, OB spread all around the world with negative consequences on health, such as a higher risk to develop cardiovascular and fatty liver diseases and Type 2 diabetes [5,6,7,8]. In fact, the World Health Organization (WHO) reported that more than 15% of the children are considered overweight and obese [9]. In Italy, 29% of the children are overweight and 10% of them are considered obese. Childhood OB has been shown to be related to an increased risk of developing several diseases such as cardiovascular, respiratory, and metabolic diseases, with negative repercussions on daily life activities [10,11,12]. Moreover, several studies reported that obese children tend to be obese even in adulthood with higher risk of morbidity and mortality [13]. To deal with childhood OB, the most effective non-pharmacological treatments are the modifications of poor health behaviors, such as the reduction of sedentary activities and increasing the daily energy expenditure. Several studies demonstrated the positive effects of engaging in regular PA practice with the reduction of childhood OB and poor health-related outcomes [14]. For example, Nemet et al. [15] showed that PA practice reduced body mass index (BMI), body fat percentage, serum total cholesterol level, and low-density lipoprotein cholesterol level. Moreover, a meta-analysis by Garcia-Hermoso et al. [16] showed that a concurrent training program is more efficient compared to aerobic training in improving health outcomes in children with obesity. Even if the benefits of PA are clear and the WHO recommends [17] at least 60 min per day of moderate activities to achieve the positive effects of PA practice on physical and psychological health, children with OB did not reach these recommendations [18]. A low PA level is related to a low level of physical fitness (PF) (defined as the ability to perform PA) [19,20,21], and PF is composed of different outcomes such as muscular strength (MS) [22], cardiorespiratory health (CRH), and speed–agility (SA). A low level of PF is considered a precursor risk of diseases development [23,24,25], and several studies showed a positive association between PF components and health-related outcomes [26,27,28]. For example, MS and SA are associated with better bone health and with a reduced risk of hypertension and Type 2 diabetes [26,27,28], while poor CRH has been related to a higher risk of cardiovascular disease morbidity and mortality [29], poorer mental health [30], cancer [31], and increased risk of all-cause mortality [23].

To improve PF in children with OB during COVID-19 restrictions, sport specialists started to deliver physical training through tele-exercise [32]. In fact, many online technologies and electronic devices were developed in these years, and training programs were enhanced through web channels, applications, and online platforms [33]. To the best of our knowledge, it is unclear the feasibility of an online training program for children with OB [33] and its efficacy in improving the PF. For these reasons, the aim of this study was to evaluate the effects of a 12-week online supervised training program in children with OB on different PF components and PA levels.

## 2. Materials and Methods

### 2.1. Participants

We considered a total of 40 Caucasian children (9 F/31 M; aged 11 ± 1.9 years) with OB (BMI z-score ≥ 2, according to World Health Organization). The children were consecutively enrolled in the Pediatric Hospital Vittore Buzzi Children’s Hospital of Milan (Italy) from March 2021 to December 2021. Subjects were referred to our institution for obesity by their general practitioner or by their primary care pediatric consultant. The patients were asked to participate in the study during a pediatric specialistic visit.

The children that participated in our study were aged between 8 and 13 years, had a BMI z-score ≥ 2, and had Italian language proficiency. Exclusion criteria were known secondary obesity conditions, non-comprehension of Italian language, cardiovascular and respiratory chronic diseases, comorbidities, orthopedic injuries, and absolute contraindications to the PA practice. The parents or guardians gave written informed consent to participate in the study, after the study protocol explanation. Children could withdraw from the study at any moment without consequences. The study protocol was approved by the Ethical Committee (protocol number 2020/ST/298).

During the examination, children performed all the physical tests to assess PF and were asked to fill out two different questionnaires: the International Fitness Enjoyment Scale (IFIS) questionnaire and the Physical Activity Questionnaire for Older Children (PAQ-C).

### 2.2. Anthropometric Characteristics

In all the subjects, height, weight, pubertal stages, BMI z-score, waist circumference (WC), and waist-to-height ratio (WHtR) were measured.

Weight was assessed standing upright in light clothing in the center of a scale platform (Seca, Hamburg, Germany) with hands at the sides looking straight ahead and facing the recorder. Standing height was measured using a Harpenden stadiometer (Holtain Ltd., Cross-well, Crymych, UK) with a fixed vertical backboard and an adjustable headpiece [34,35]. WC was measured using a flexible inch tape, in the horizontal plane midway between the lowest ribs and the iliac crest [34,35]. The measurement was taken on the child in an upright position, without shoes, with their heels together and toes apart, hands at sides, aligning the head in the Frankfort horizontal plane [36]. BMI was calculated as body weight (kilograms) divided by height (meters squared), and was transformed into BMI z-scores using WHO reference values [37]. Pubertal stages according to Tanner were classified as follows: prepubertal stage 1 = Tanner 1; middle puberty stage 2 = Tanner 2–3; late puberty stage 3 = Tanner 4–5 [38,39].

### 2.3. The Physical Activity Questionnaire for Older Children (PAQ-C)

The PAQ-C is a self-reported questionnaire that evaluates the weekly amount of physical activity.

This questionnaire was proven to be appropriate for school-aged children (approximately ages 8–14) who are currently in the school system. The PAQ-C is supported as a valid and reliable measure of general physical activity levels from childhood to adolescence [40]. The PAQ-C utilizes memory cues such as recess time at school and physical activity performed in the evening to improve the recall ability of children. The PAQ-C is cost- and time-efficient, easy to administer to large-scale populations, and displays normal distribution properties. The PAQ-C is shown to have good reliability and an intraclass correlation (ICC) = 0.96 [40].

### 2.4. Children’s Effort Rating Table

The Children’s Effort Rating Table (CERT) is a 0–10-point scale [41] that allows to assess the perceived fatigue in children after and during an exercise. Previous studies showed its validity and reliability in children (ICC = 0.91) [41], and it had greater validity in children (aged 8–11) compared to the Borg 6–20 Rating of Perceived Exertion Scale.

### 2.5. Physical Fitness Tests

The data collection consisted of a series of PF tests [42,43]. These field tests are valid and reliable tools for measuring PF in children and are widespread, inexpensive as equipment, and easy to administer [44,45].

### 2.6. Standing Broad Jump (SBJ)

The SBJ is a reliable and valid method to assess the strength and power of the lower limbs [22,46]. Before the assessment, trainers explained the procedure and showed how to perform the test. Each child started from a standing position placing both feet behind the starting line. After preparatory movements, a horizontal jump was performed with the contribution of the upper limbs in free swing. The distance (to the nearest 0.5 cm) from the starting line to the heel of the rear foot was recorded. The test was performed two times, with a five-minute rest between each attempt, and the best score was retained for investigation. The standing broad jump presents high reliability from test–retest analysis, and the ICCs reported a range from 0.94 to 0.95 [46].

### 2.7. 6 Min Walking Test (6MWT)

The 6MWT was performed according to international guidelines [47]. The children were instructed by the trainers to walk the greatest distance possible while maintaining their own pace. Every minute, standardized encouragement and information about the remaining time were given to the children, for example, “You are doing well” or “Keep up the good work” [48,49].

Patients were permitted to stop (if required) during the test but were instructed to resume walking once able. The covered distance was registered in meters whilst perceived fatigue was assessed with the Children’s Effort Rating Table (CERT) scale, a 0–10-point scale, which has been shown to be a reliable scale to assess perceived fatigue in children. Test–retest reliability was undertaken, and the intraclass correlation coefficient (95% confidence interval) was calculated as 0.94 (0.89–0.96) [50].

### 2.8. 5 × 10 m Sprint Test

This test is part of the Eurofit fitness test battery [42] and is commonly used to assess speed agility. It consists of running and turning as fast as possible between two parallel lines placed 10 m apart five times. The trainers first explained to the children how to perform the test and then showed them how to perform it. The time was taken using a chronograph to record the time (stopwatch W073, SEIKO, Tokyo, Japan), and a lower time indicated better performance. The 5 × 10 m shuttle run test presents high reliability, and the ICC was calculated as 0.95 [51].

### 2.9. Training Protocol

The participants were asked to participate in a 60 min exercise program on the online platform Zoom. The training protocol consisted of three 60 min sessions (on Mondays, Wednesdays, and Fridays) per week over 12 weeks for a total of 36 sessions. A 12-week protocol was chosen because several studies reported that 12 weeks are sufficient to improve both physical fitness and health outcomes in children with obesity [52,53,54,55]. Each session was streamed in real time using the Zoom platform and allowed live interaction among the instructors and participants. Two trainers supervised every training session which, usually, consisted of 5 min of warm-up, 50 min of a combination of aerobic and strength exercises, and ended with 5 min of stretching. All the exercises proposed were playful activities and did not require any specific equipment. All the training sessions were composed of a 5–10 min warm-up, 20 min of aerobic interval training, 20 min of strength circuit, and 5–10 min of cool-down [56]. Table 1 shows an example of a typical training session. To understand the intensity of the training session, the heart rate was monitored with an activity tracker (Fitbit Charge 2^©^, Fitbit Inc., San Francisco, CA, USA) and registered by the trainers after 30 min from the beginning of the sessions. Moreover, before each exercise, trainers reminded children to maintain an intensity based on a CERT scale value, as shown in Table 1. Both training intensity and duration were kept constant throughout the 12 weeks of training. The study protocol is presented in Figure 1.

### 2.10. Statistical Analysis

All quantitative data were summarized as mean and standard deviation (SD). We tested for normality by Shapiro–Wilk tests and graphically checked for linearity. We used a nonparametric paired samples Wilcoxon W *t*-test to evaluate the differences before and after the training period for the PAQ-C scores, WHtR, and the CERT scale. The rank biserial correlation was used to estimate the size of the effect, while a parametric paired sample Student’s *t*-test was used to assess the difference between the BMI z-score, WC, and PF tests before and after the training period, and Cohen’s d to assess the size of the effect. All the significance was set at a *p*-value less than 0.05. Statistical analyses were performed using The Jamovi Project (2021). Jamovi Version 1.6 for Mac [Computer Software], Sydney, Australia; retrieved from https://www.jamovi.org (accessed on 5 May 2022).

## 3. Results

A total of 37 patients (28 M/ 9 F, aged 11 + 1.90 years; pubertal stage 1 = 21; pubertal stage 2 = 8; pubertal stage 3 = 8) completed the training protocol and were considered in the analysis. The anthropometric data of the children are shown in Table 2.

As reported in Table 2, after the training protocol a significant decrease in the BMI z-score (*p* = *0*.006), WC (*p* = *0*.027), and WHtR (*p* = *0*.005) were recorded.

The PF test performances of the children before and after the training are reported in Table 3.

For the SBJ performance, there was a significant increase in the distance jumped by the children, of 10.00 cm, after the 12 weeks of training (*p* < 0.001). In addition, the distance covered during the 6MWT significantly increased after the training, with a mean difference of 54.92 m (*p* < 0.001). The 5 × 10 m sprint test analysis showed a significant decrease in the test time with a mean difference of 0.98 s after the training (*p* = 0.014).

No significant correlation between changes in the PF and auxological parameters were noted except for SBJ and BMI z-score (*p* = 0.03) and WC/H (*p* = 0.03).

Table 4 shows the data of the CERT and PAQ-C scores before and after the 12 weeks of intervention.

The analysis showed that the CERT scale’s scores did not significantly change both in the 6MWT and in the 5 × 10 sprint test (*p* > 0.05). On the contrary, the scores of the PAQ-C had a slight increment after the training (*p* = 0.05).

## 4. Discussion

During the COVID-19 pandemic, many online programs helped people to exercise during the restriction imposed to protect people to the infection [33,57,58,59], but there is limited knowledge on online exercise adapted to children with OB. To the best of the authors’ knowledge, this is the first Italian study to investigate the effects on body mass, physical activity level, and PF of 12 weeks of supervised online training in a sedentary, healthy, youth population with obesity. Hence, with these preliminary results, we highlight the importance to develop online training to maintain functional capacity and control weight gain in youth with OB.

Childhood obesity is a multisystem disease with negative consequences and various comorbidities, thereby contributing to premature death in adulthood [60]. Physical exercise represents a non-pharmacological intervention crucial to mitigate the problems of overweight and obesity, as well as related comorbidities starting from childhood [61]; several studies demonstrated the importance of the reduction of the BMI z-score, WC, and WHtR in children with OB to prevent and/or reduce the cardiovascular risk and to obtain better health outcomes [62,63,64,65,66,67,68,69].

Our study showed that 12 weeks of an online supervised training program can reduce the BMI z-score, the WC, and the WHtR in children with OB. In accordance with our results, Pancar et al. [70] showed a reduction of the BMI with 4 weeks of low-intensity PA in children with OB, while Bethea et al. [71] found a reduction in the BMI z-score, glucose, and total cholesterol with 30 weeks of online exergames intervention. Furthermore, Tan et al. [72] showed a reduction in the WC with a 10-week face-to-face exercise program in children with OB. Moreover Pienaar et al. [73], with 4 weeks of PA and diet intervention, showed a significant reduction in the BMI, body fat percentage, and skinfold circumferences in 11 years old children with OB. On the contrary, Romero-Perez did not find any significant changes in the BMI z-score with a 20-week recreative and aerobic training program in children and adolescents with OB [74]. Although several studies analyzed the effects of promoting face-to-face or online unsupervised PA intervention on anthropometric outcomes, our study is the first to investigate the effect of an online supervised training program on body composition parameters in children with OB.

Moreover, we found that an online supervised training program was effective in improving PF. Specifically, we found that the training protocol strongly increased cardiorespiratory fitness (*p* < 0.001; *effect size* = 1.08), muscular strength, and lower limb power (*p <* 0.001; *effect size* = 0.853), while there was an adequate improvement in the speed–agility (*p* = 0.014; *effect size* = 0.496). To the best of our knowledge, there is a lack of data regarding the effectiveness of online supervised training programs on the improvement of the PF in children with OB. Only one study, by Lee [75], showed the effectiveness of online supervised training in improving the MS in middle school children. Despite this lack of data, several studies showed the effectiveness of a supervised training program for enhancing the PF in children with OB. For example, Molina-Garcia et al. [76] found that 13 weeks of face-to-face training can improve the performance of the SBJ and the 1RM arm and handgrip strength test in children with OB. It is important to underline that in our study the children after the PF test did not report any significant changes in their effort (CERT 6MWT *p* > 0.05 and CERT 5 × 10 m *p* > 0.05), even if their performance after the training protocol was improved, showing that with the same effort they had better performance in the same test.

The effectiveness of the online supervised training program in our study is strengthened by the data of the PAQ-C that showed a significant increase in the weekly level of PA after the training protocol. In fact, as several research demonstrated, a high level of PA is associated with a better health-related quality of life in children and adolescents [77,78]. Unfortunately, even if an improvement in the weekly PA was observed with our training program, only three children after the training program declared to reach the cut-off value of 2.73 in PAQ_C [79], which represents the amount of recommended PA by WHO guidelines to achieve health benefits [17].

We recognize that our study had some limitations. The number of participants is limited and, in the future, an increased sample size is expected to extend and validate these first results. Secondly, we did not have any control group to understand if the same improvement could be obtained even with face-to-face training, and, finally, we did not analyze any specific metabolic and medical indicators associated with OB.

Nevertheless, our study, with a reduced dropout rate (less than 10% of the initial sample), demonstrated the feasibility of an online supervised training program for children with OB, probably thanks to an adapted exercise program and the absence of a competitive context. Future studies should evaluate the effectiveness of an online training program in other populations and confirm its efficacy in children with OB by analyzing specific metabolic and other medical data associated with OB.

## 5. Conclusions

In conclusion, the results of our study showed that an online supervised training program is effective to promote PA, improving PF and reducing the BMI z-score in children with OB. Therefore, our study should be used as a starting point for a new way to promote exercise and PA in children with OB, through an online platform, in order to increase the PA adherence and promote an active lifestyle in children with OB.

## Figures and Tables

**Figure 1 ijerph-19-09421-f001:**
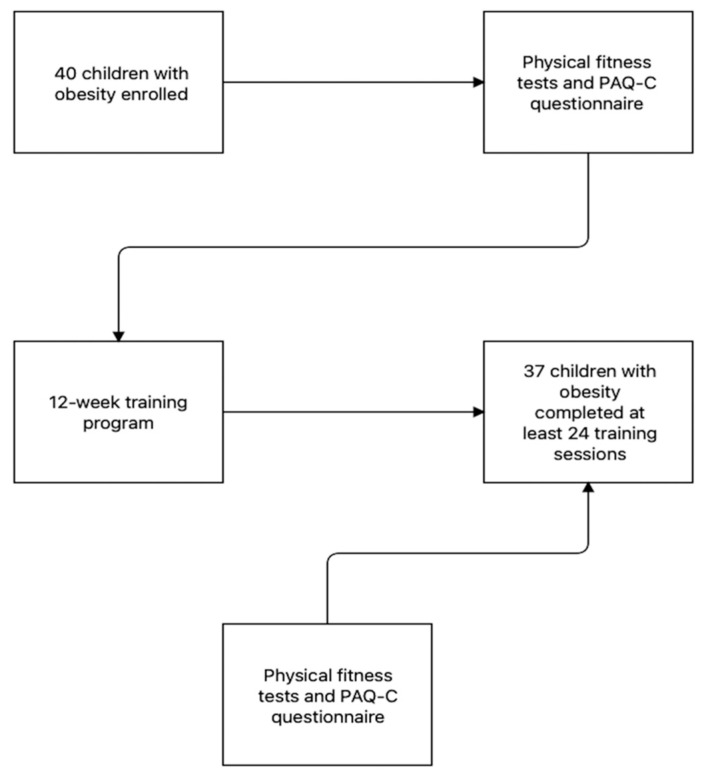
Flow chart of the study protocol.

**Table 1 ijerph-19-09421-t001:** Example of a typical training session.

	Warm-Up (5–10 min)	Aerobic Interval Training (20 min)	Muscular Strength (20 min)	Cool-Down (5–10 min)
**Type of exercises**	**Mobility exercises** (e.g.,. head, lower and upper body mobility routine)	**Motor tales** (e.g., “a journey through the woods”, children imagined being in the woods completing different tasks, which included different aerobic activities (run, jumping, march …)	**Strength circuit for children** (e.g., semi-squat, push-up, glute bridge, triceps dip … all exercises were performed free weight)	**Yoga for children** (e.g., child pose, reverse warrior, cobra pose, seated forward fold, corpse pose …)
	**Active music video** (the music video was performed by children imitating the dancers)	**Animal walks** (e.g., bear crawl, frog jump, bunny hop, dinosaur walk)	**Imitation games** (e.g., imitation of multiple sport techniques and gestures)	
**Intensity**	4–5 CERT	7–8 CERT	7–8 CERT	3–4 CERT

Note: CERT = Children’s Effort Rating Table.

**Table 2 ijerph-19-09421-t002:** Anthropometric data of the sample.

	Before TP	After TP	*p*-Value
**Weight** **-kilograms (kg)** **-percentiles ^§^**	65.74 ±17.88 97.18 ± 3.58	65.80 ± 15.77 97.20 ± 4.0	0.78 0.96
**Height** **-meters** **-percentiles ^§^**	1.49 ± 0.10 50.90 ± 31.38	1.51 ± 0.11 59.70 ± 32.06	0.36 0.28
**Waist Circumference (cm)**	90.54 ± 11.21	88.22 ±10.86	0.027 *
**WHtR**	0.59 ± 0.13	0.56 ± 0.12	0.005 *
**BMI**	28.74 ± 4.10	28.50 ± 3.55	>0.05
**-BMI z-score** **-percentiles ^§^**	2.20 ± 0.22 99.37 ± 0.89	2.16 ± 0.25 98.89 ± 1.86	0.006 ** 0.19

Note: * *p* < 0.05; ** *p* < 0.01. cm = centimeters; kg = kilograms; WHtR = waist-to-height ratio. Pubertal stages are represented as frequencies; waist circumferences, WHtR, and BMI z-score as mean + SD. ^§^ According to the World Health Organization.

**Table 3 ijerph-19-09421-t003:** Physical fitness test results before and after the training program.

	Before	After	*p*-Value	*Effect Size*
**SBJ (cm)**	99.29 ± 20.16	109.29 ± 22.84	<0.001 ***	0.853
**6MWT (m)**	479.50 ± 61.11	534 ± 61.08	<0.001 ***	1.08
**5 × 10 m (s)**	19.91 ± 2.45	18.93 ± 1.91	0.014 *	0.496

Note: * *p* < 0.05; *** *p* < 0.001; SBJ = standing broad jump; 6MWT = 6 Minute Walking Test; cm = centimeters; m = meters; s = seconds.

**Table 4 ijerph-19-09421-t004:** CERT and PAQ-C scores before and after the training program.

	Before	After	*p*-Value	*Effect Size*
**CERT 6MWT**	4.00 (3.00–6.00; 3)	5.00 (2.75–6.25; 3.50)	0.638	0.080
**CERT 5 × 10 m**	5.00 (4.00–7.00; 3.00)	5.00 (4.00–7.00; 3.00)	0.245	0.184
**PAQ-C**	1.97 (1.65–2.24; 0.585)	2.27 (1.78–2.59; 0.810)	0.050 *	0.400

Note: * *p* < *0*.05; 6MWT = 6 Minute Walking Test; CERT = Children’s Effort Rating Table; PAQ-C= Physical Activity Questionnaire for Older Children.

## Data Availability

The data presented in this study are available on request from the corresponding author. The data are not publicly available due to privacy reasons.

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
