# Peer review of "Effects of an Online Supervised Exercise Training in Children with Obesity during the COVID-19 Pandemic"

_ijerph, 2022, doi:10.3390/ijerph19159421_

Round 1

Reviewer 1 Report

This is a very interesting study and is well written. There are some places to be checked and corrected.

Line 88: Even though the authors determined the BMI z-score, the height, the weight, and the BMI data should be provided too.

Line 125-126: Please put the reference about the PAQ-C.

Line 133: Please put the reference about the CERT.

Line 142: Please put the reference about the SBJ.

Line 149: Put the reference number [44] to the end of the sentence.

Line 171: The training protocol was effective so that it should be explained clearly. I suggest adding a Table showing it.

Table 1: “p”-value or “p”? Please unify.

Table 2: “Note: * p<. “ ---The beginning of the sentence is bold. Please fix.

Table 3: “Note: * p<.05; ** p<.01; *** p<.001 6MWT=6-minute” ---The font was different.

Line 278: The authors should provide the participant numbers and the dropping out numbers in the Methods.

Author Response

This is a very interesting study and is well written. There are some places to be checked and corrected.

R: Thanks for your precious work that surely ameliorated the full body of the manuscript. We appreciate the time that you dedicated to the revision of our work. 

Line 88: Even though the authors determined the BMI z-score, the height, the weight, and the BMI data should be provided too.

R: Thank you for your valuable suggestion, we added the BMI data in table 2.

Line 125-126: Please put the reference about the PAQ-C.

Line 133: Please put the reference about the CERT.

Line 142: Please put the reference about the SBJ.

Line 149: Put the reference number [44] to the end of the sentence.

R: Thank you for your valuable suggestions, we added all the references in the text.

Line 171: The training protocol was effective so that it should be explained clearly. I suggest adding a Table showing it.

R: Thanks for the useful suggestion: we modified the title according to your considerations.

Table 1: “p”-value or “p”? Please unify.

R: We unified it according to your suggestion.

Table 2: “Note: * p<. “ ---The beginning of the sentence is bold. Please fix.

R: Thank you for your precious comment, we fixed it.

Table 3: “Note: * p<.05; ** p<.01; *** p<.001 6MWT=6-minute” ---The font was different.

R: Thank you for pointing it out, we modified it.

Line 278: The authors should provide the participant numbers and the dropping out numbers in the Methods.

R: Thank you for your valuable suggestion, we provided the number of the enrolled children in the method section (Line 88) and the number of the children that completed the study in the results section (Line 196)

Reviewer 2 Report

Many thanks to the editors of the journal for considering me to review this work and thanks to the authors for the time they have dedicated to the preparation of this interesting paper.
First of all, it should be pointed out that the subject matter is relevant and of current interest to scientists, since it deals with the current pandemic we are experiencing and the performance of physical exercise at a distance. Methodology of work that will continue to be present.
Entering the content of the paper, it can be seen in the abstract takes into account the main points that should appear such as the objectives, brief description of the sample, type of statistical analysis, results and conclusion.
As for the introduction, it can be seen that the authors focus on childhood obesity, on the problems it entails (which has been dealt with extensively in other studies and may be dispensable in this one) and on the evolution throughout the pandemic. At one point in the introduction (line 70) the authors focus on Physical Fitness and it seems to be a variable on a par with childhood obesity. Evidently, they have a direct relationship, but it should be clear in the introduction the importance they give to this variable since later in line 78 they put the focus of the paragraph of the objective on this variable and, the training program that marks the objective, has the purpose of increasing the PF to reduce childhood obesity.
Regarding the methodology, they start with the participants and it should be pointed out that they make a good description and that it is a population that is difficult to access, so a sample of 40 is considered optimal. On the other hand, there is an adequate description of the measurement instruments and statistical tests.
The results are clear and focused on the objective and the title and therefore meet the expectations. In order to improve the presentation, it is requested that the title of "results" remain on the next page, Table 3 is not cut off.
In the discussion, they highlight the most important findings of this study and compare them with similar studies. It is true that the discussion regarding the FP results and the results of the PAQ-C questionnaire are superficial and should be improved.
In the conclusions, it is considered that should be extracted from the results and not establish interpretations or recommendations for other studies, such as the conclusion that goes from lines 285 to 287

Author Response

Many thanks to the editors of the journal for considering me to review this work and thanks to the authors for the time they have dedicated to the preparation of this interesting paper.
First of all, it should be pointed out that the subject matter is relevant and of current interest to scientists, since it deals with the current pandemic we are experiencing and the performance of physical exercise at a distance. Methodology of work that will continue to be present. 
Entering the content of the paper, it can be seen in the abstract takes into account the main points that should appear such as the objectives, brief description of the sample, type of statistical analysis, results and conclusion. 
As for the introduction, it can be seen that the authors focus on childhood obesity, on the problems it entails (which has been dealt with extensively in other studies and may be dispensable in this one) and on the evolution throughout the pandemic. At one point in the introduction (line 70) the authors focus on Physical Fitness and it seems to be a variable on a par with childhood obesity. Evidently, they have a direct relationship, but it should be clear in the introduction the importance they give to this variable since later in line 78 they put the focus of the paragraph of the objective on this variable and, the training program that marks the objective, has the purpose of increasing the PF to reduce childhood obesity. 
Regarding the methodology, they start with the participants and it should be pointed out that they make a good description and that it is a population that is difficult to access, so a sample of 40 is considered optimal. On the other hand, there is an adequate description of the measurement instruments and statistical tests. 
The results are clear and focused on the objective and the title and therefore meet the expectations. In order to improve the presentation, it is requested that the title of "results" remain on the next page, Table 3 is not cut off.
In the discussion, they highlight the most important findings of this study and compare them with similar studies. It is true that the discussion regarding the FP results and the results of the PAQ-C questionnaire are superficial and should be improved.

R: Thanks for your precious work that surely ameliorated the full body of the manuscript. We appreciate a lot the time that you dedicated to the revision of our work. We improved the discussion about the PF tests and the PAQ-C results following your valuable suggestion. 

In the conclusions, it is considered that should be extracted from the results and not establish interpretations or recommendations for other studies, such as the conclusion that goes from lines 285 to 287

R: Thank you for the precious comment, we moved the last sentences of the conclusions paragraph in the discussion.

Reviewer 3 Report

Dear authors, please find below some comments on your manuscript.

In the Introduction please provide the information on which type of exercise is the most recommended for obese children.
Was the study protocol registered in any database?
Please provide a flow chart of the study.
Were there any subject discontinuation criteria from the study? How many training participants should complete to be included in the analysis?
Did the children complete all questionnaires alone or with the help of their parents?
Did the participants have to have cameras turned on during the exercises?
In the Materials ad Methods please provide more details about the training programme. On what days of the week were the training performed? Any equipment was needed during the training? What exactly types of strength and aerobic exercises were performed? What was the intensity of training? Did the intensity and duration of training change throughout the intervention? Did the participants receive any guidelines about heart rate that should be maintained during the training? Why the duration of the intervention was 12 weeks?
Did the authors assess the dietary habits of study participants?
Did the authors calculate the minimum sample size?
Any adverse effects of the intervention were reported?
Please present the characteristics of the study population in the table.
Why did the authors not present in Table 1 weight and height?
Please consider presenting in Table 1 waist circumference as a percentile.
In the sentence „Although several studies analyzed the effects of promoting face-to-face or online unsupervised PA intervention on body composition parameters, our study is the first that investigated the effect of an online supervised training program on body composition parameters in children with OB.” please replace „body composition” by „anthropometric parameters”.
In the Funding section please provide the source of APC founding.
Why are the data from the study not publicly available (see Data Availability Statement)?
Please remove the duplicate numbering of papers included in the bibliography.

Author Response

Dear authors, please find below some comments on your manuscript.

R: Thanks for your precious work that surely ameliorated the full body of the manuscript. We appreciate the time that you dedicated to the revision of our work. 

In the Introduction please provide the information on which type of exercise is the most recommended for obese children.

R: Thank you for your valuable suggestion, we added it in lines 56-58

Was the study protocol registered in any database?

R: No, the study was not registered in any database.

Please provide a flow chart of the study.

R: Thank you for the precious comment, we added it.

Were there any subject discontinuation criteria from the study? How many training participants should complete to be included in the analysis?

R: We did not have any discontinuation criteria and participants must have completed at least 24 training sessions out of the possible 36 to be included in the analyses.

Did the children complete all questionnaires alone or with the help of their parents?

R: The children filled out the questionnaire on their own driven by investigators.

Did the participants have to have cameras turned on during the exercises?

R: Yes, before the training start, instructors checked that all participants had their cameras on.

In the Materials ad Methods please provide more details about the training programme. On what days of the week were the training performed? Any equipment was needed during the training? What exactly types of strength and aerobic exercises were performed? What was the intensity of training? Did the intensity and duration of training change throughout the intervention? Did the participants receive any guidelines about heart rate that should be maintained during the training? Why the duration of the intervention was 12 weeks?

R: Thanks for the useful suggestion, we added them and we created a table to better explain our training program (table 1).

Did the authors assess the dietary habits of study participants?

R: We assessed the dietary habits as a clinical routine, however, since no change of habits was required, these data were not analyzed.

Did the authors calculate the minimum sample size?

R: Yes, we calculated the sample size considering the possible blood pressure outcomes changes that are not part of the present in this study, for this reason, we did not add the sample size calculation in this study.

Any adverse effects of the intervention were reported?

R: No, we did not report any adverse effects.

Please present the characteristics of the study population in the table.

R: Thank you for the precious comment, we added the descriptive characteristics in table 1.

Why did the authors not present in Table 1 weight and height

R: We thought that presenting another table in the text could weigh down the structure of the table, so we decided to put the mean and standard deviation values in the text (lines 244-246). Despite this, we added weight and height in table 1.

Please consider presenting in Table 1 waist circumference as a percentile.

R: Thank you for the precious suggestion, unfortunately, we did not calculate them.

In the sentence „Although several studies analyzed the effects of promoting face-to-face or online unsupervised PA intervention on body composition parameters, our study is the first that investigated the effect of an online supervised training program on body composition parameters in children with OB.” please replace „body composition” by „anthropometric parameters”.

R: Thank you for your valuable suggestion, we modified it.

In the Funding section please provide the source of APC founding.

R: We did not pay any money for the APC because the corresponding author is one of the editors of the special issue.

Why are the data from the study not publicly available (see Data Availability Statement)?

R: The data presented in this study are available on request from the corresponding author. The data are not publicly available due to privacy reasons.

Please remove the duplicate numbering of papers included in the bibliography.

R: We are sorry, but we can not find the duplicate numbering of papers in the biography

Round 2

Reviewer 3 Report

In response, the authors mentioned that "participants must have completed at least 24 training sessions out of the possible 36 to be included in the analyses". However, according to Figure 1, it was enough to complete 12 training to be included in the analysis. Is there a mistake in the Figure?
Did the authors analyze how the adherence to the intervention impacted the results?
Please consider providing weight and height also as percentiles.
What does "//" mean in Table 2?'
Why did the authors use to calculate the minimum sample size the outcome that is not presented in the manuscript?

Author Response

In response, the authors mentioned that "participants must have completed at least 24 training sessions out of the possible 36 to be included in the analyses". However, according to Figure 1, it was enough to complete 12 training to be included in the analysis. Is there a mistake in the Figure?

We are sorry, in the figure there is a typo..thanks to report it

Did the authors analyze how the adherence to the intervention impacted the results? 

It is a good suggestion but actually we did not analyze it.

Please consider providing weight and height also as percentiles.

We added the values in the text

What does "//" mean in Table 2?' 

It means that we did not calculate the p-values. We added the note under the table.

Why did the authors use to calculate the minimum sample size the outcome that is not presented in the manuscript?

Because this results are part of a larger study that has as main outcame the blood pressure